# MAI: A Multi-turn Aggregation-Iteration Model for Composed Image Retrieval

**Yanzhe Chen**[1], **Zhiwen Yang**[1], **Jinglin Xu**[2], **Yuxin Peng**[1]*

[1]Wangxuan Institute of Computer Technology, Peking University
[2]School of Intelligence Science and Technology, University of Science and Technology Beijing
`chenyanzhe@stu.pku.edu.cn, yangzhiwen@pku.edu.cn,`
`xujinglinlove@gmail.com, pengyuxin@pku.edu.cn`

## Abstract

Multi-Turn Composed Image Retrieval (MTCIR) addresses a real-world scenario where users iteratively refine retrieval results by providing additional information until a target meeting all their requirements is found. Existing methods primarily achieve MTCIR through a "multiple single-turn" paradigm, wherein methods incorrectly converge on shortcuts that only utilize the most recent turn's image, ignoring attributes from historical turns. Consequently, retrieval failures occur when modification requests involve historical information. We argue that explicitly incorporating historical information into the modified text is crucial to addressing this issue. To this end, we build a new retrospective-based MTCIR dataset, **FashionMT**, wherein modification demands are highly associated with historical turns. We also propose a Multi-turn Aggregation-Iteration (**MAI**) model, emphasizing efficient aggregation of multimodal semantics and optimization of information propagation in multi-turn retrieval. Specifically, we propose a new Two-stage Semantic Aggregation (TSA) paradigm coupled with a Cyclic Combination Loss (CCL), achieving improved semantic consistency and modality alignment by progressively interacting the reference image with its caption and the modified text. In addition, we design a Multi-turn Iterative Optimization (MIO) mechanism that dynamically selects representative tokens and reduces redundancy during multi-turn iterations. Extensive experiments demonstrate that the proposed MAI model achieves substantial improvements over state-of-the-art methods. The dataset and source code are available at https://github.com/PKU-ICST-MIPL/MAI_ICLR2025.

## 1 Introduction

Image retrieval remains a longstanding task in computer vision Sain et al. (2023); Levy et al. (2024a), gaining continuous attention in practical applications such as e-commerce in recent years Jin et al. (2023); Park et al. (2019). However, relying solely on images may fall short of practical needs, as users often modify these images better to match their requirements Chen et al. (2020); Guo et al. (2018). In response, Composed Image Retrieval (CIR) has been introduced to locate target images by combining reference images and modified text Wen et al. (2023); Shoib et al. (2023). Due to the interactive nature of retrieval scenario Xu & Sundar (2014); Adhikari et al. (2018), multi-turn systems can leverage more user feedback, fulfilling user needs better than single-turn systems Agnolucci et al. (2023); Chen et al. (2023a). Therefore, Multi-turn Composed Image Retrieval (MT-CIR), which aims to retrieve the most suitable target image by allowing users to iteratively select images and provide modification feedback, as illustrated in Figure 1, has garnered increasing attention in recent years Guo et al. (2018); Yuan & Lam (2021); Liu et al. (2024b).

Due to the lack of dedicated datasets for the MTCIR task Pal et al. (2023), existing methods typically construct multi-turn datasets by concatenating single-turn CIR datasets Wu et al. (2021); Guo et al. (2018), using the target image from the historical turn as the reference image for the next turn. However, datasets constructed in this manner exhibit the following limitations: **(i) Lack of historical**

---

*Corresponding author.

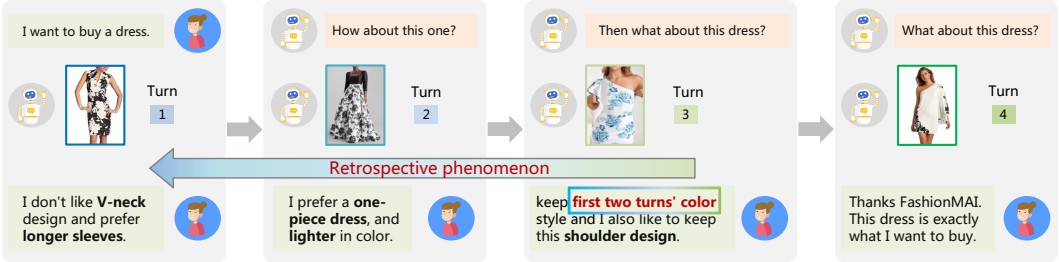

Figure 1: The definition of Multi-Turn Composed Image Retrieval (**MTCIR**). The retrospective phenomenon is common in the MTCIR task, wherein a user's new turn modification request often involves the attributes of images from historical turns.

**context.** Modified text in existing MTCIR datasets lacks image information from historical turns, resembling "multiple single-turn" retrievals and deviating from real-world scenarios. **(ii) Small data scale.** Existing single-turn datasets face challenges due to their limited scale Saito et al. (2023); Feng et al. (2024); Zhao et al. (2024b). Moreover, this concatenation method further diminishes the size of multi-turn datasets, lagging behind current trends Chen et al. (2023b); Baldrati et al. (2023).

The deficiencies outlined in existing MTCIR datasets have hindered the development of methods in this domain. Existing methods typically employ a "multiple single-turn" paradigm for multi-turn retrieval. However, this paradigm causes methods to incorrectly converge on shortcuts that only utilize the most recent turn's image, neglecting attributes from previous turns. Consequently, retrieval failures arise when modification requests involve attributes or modifications from previous images. Additionally, existing methods lack designs to leverage inherent multimodal information in images Chen et al. (2023b); Li et al. (2024a), and to store multi-turn information effectively.

To address these issues and align with existing MTCIR datasets, we construct a new dataset, **FashionMT**, tailored for e-commerce scenarios characterized by typical multi-turn interactions. FashionMT has the following characteristics: **(i) Retrospective-based.** It simulates real-world MTCIR scenarios, where the modified text in each new turn may involve information from historical reference images, such as preserving certain attributes. This necessitates retrieval algorithms to utilize multi-turn historical information retrospectively. **(ii) Massive and diverse.** FashionMT contains 14 times more fashion images and 30 times more categories than MT FashionIQ Yuan & Lam (2021). Our Modification Generation Framework generates multi-turn transactions nearly 27 times larger than MT FashionIQ, offering rich multimodal data, including images, text, attributes, etc.

We further propose a multi-turn key information-aware approach, the Multi-turn Aggregation-Iteration (**MAI**) model, which focuses on two challenges in MTCIR: **(i) multimodal semantics aggregation** and **(ii) multi-turn information optimization.** Specifically, MAI introduces a new Two-stage Semantic Aggregation (TSA) paradigm coupled with a Cyclic Combination Loss (CCL). TSA introduces captions as a transition, progressively aggregating the image with its caption and then with the modified text. The CCL's cyclic structure further enhances semantic consistency and modality alignment. We also provide theoretical insights into the rationale behind introducing captions for two-stage fusion. Furthermore, we design a parameter-free Multi-turn Iterative Optimization (MIO) mechanism that dynamically selects representative tokens with high semantic diversity, effectively reducing the storage space for historical information tokens.

Our contributions are summarized as follows:

- We build the first dataset specifically designed for multi-turn composed image retrieval, named FashionMT, characterized by its retrospective-based nature and massive diversity.

- We propose the Multi-turn Aggregation-Iteration (MAI) model, focusing on efficient aggregation and iterative optimization of multimodal semantics in multi-turn composed image retrieval.

- We provide theoretical insights that our modality fusion approach effectively bridges the modality and semantic gaps, which informs the design of our loss function.

- Extensive experiments demonstrate that our proposed MAI model obtains substantial improvements and achieves state-of-the-art performance.

## 2 RELATED WORK

**Single-turn Composed Image Retrieval.** In existing works on composed image retrieval, the focus has mainly been on single-turn retrieval Wen et al. (2023); Chen et al. (2024c), which can be categorized based on the amount of training data into fully trained on all data Goenka et al. (2022); Levy et al. (2024b) and zero-shot Karthik et al. (2024); Gu et al. (2023); Chen & Lai (2023) or few-shot Wu et al. (2023) settings. Currently, composed image retrieval methods can be broadly categorized into two paradigms Bai et al. (2024): late fusion Chen et al. (2024b); Zhang et al. (2024) or pseudo-word embedding methods Baldrati et al. (2023); Liu et al. (2024b); Suo et al. (2024). In the first paradigm type, Baldrati et al. (2022) propose a simple yet effective fusion model, Combiner, to combine features extracted by the CLIP Radford et al. (2021) model. In the second paradigm type, Saito et al. (2023) propose an LLAVA-like Liu et al. (2024a) method to convert visual features into tokens for a text encoder. Bai et al. (2024) propose a method similar to the BLIP-2 Li et al. (2023) to learn sentence-level prompts, achieving state-of-the-art results. However, the above methods are limited to single-turn retrieval scenarios and are challenging to apply directly to the more user-demand-oriented MTCIR tasks.

**MTCIR Methods.** Several recent methods have emerged in the fusion of visual and textual inputs across multiple exchanges of information Zhu et al. (2024); Li et al. (2024b); Hu et al. (2024). Due to the inherent multi-turn nature of dialogues Zhang et al. (2022); Yu et al. (2019), a common application scenario is multi-turn dialogue systems Zolkepli et al. (2024); Zheng et al. (2022). In the prevalent retrieval tasks of the fashion domain, multi-turn retrieval has emerged as a more comprehensive approach compared to single-turn retrieval, offering enhanced user interaction and feedback to better cater to user needs Zhang et al. (2019); Agnolucci et al. (2023). There have been several groundbreaking studies in multi-turn composed image retrieval in recent years. Guo et al. (2018) propose modeling images and text using CNN networks, capturing sequential information with RNNs, and employing reinforcement learning for constraint. Yuan & Lam (2021) construct the first multi-turn composed retrieval dataset based on the single-turn retrieval dataset FashionIQ Wu et al. (2021). Pal et al. (2023) introduce a memory network to retain historical retrieval information and further develop a multi-turn retrieval dataset based on the single-turn dataset Shoes Guo et al. (2018). However, the above methods fail to leverage the multimodal content naturally present in fashion images, such as captions and titles. Additionally, these methods do not consider optimizing the storage overhead of multi-turn representations.

**Fashion Datasets.** In the past few years, a large number of datasets have been proposed for retrieval Corbiere et al. (2017); Ge et al. (2019); Rostamzadeh et al. (2018); Han et al. (2017); Zhan et al. (2021). Due to the inherent inclusion of a vast amount of data and extensive user interactions in the e-commerce domain, existing fashion datasets exhibit a large scale. The Product1M Zhan et al. (2021) contains 1,182,083 cosmetic samples. The M5Product Dong et al. (2022) encompasses 6,131,064 samples with 5 modalities. A massive amount of data also contributes to the model acquiring capabilities closer to practical usage Chen et al. (2023b). In the composed image retrieval task, FashionIQ Wu et al. (2021) and Shoes Guo et al. (2018) represent pioneering works, being more user-friendly compared to direct image retrieval. However, the MTCIR task still lacks a dedicated custom dataset. Constructing modifications by concatenating single-turn datasets fails to capture the historical context crucial for multi-turn scenarios. Our proposed FashionMT offers essential and timely data support to advance this task.

## 3 THE FASHIONMT DATASET

### 3.1 DATA COLLECTIONS AND CONSTRUCTION

Our data primarily originates from two sources: (i) Gathering images and associated text from existing single-turn composed image retrieval datasets Wu et al. (2021); Guo et al. (2018); Han et al. (2017). (ii) Crawling images and related text from multiple e-commerce platforms. We clean the scraped images, including removing damaged, unclear, and non-product images.

Inspired by the manual annotation process of modified text Wu et al. (2021); Liu et al. (2021), we propose a Modification Generation Framework (MGF) to automate the construction of our dataset

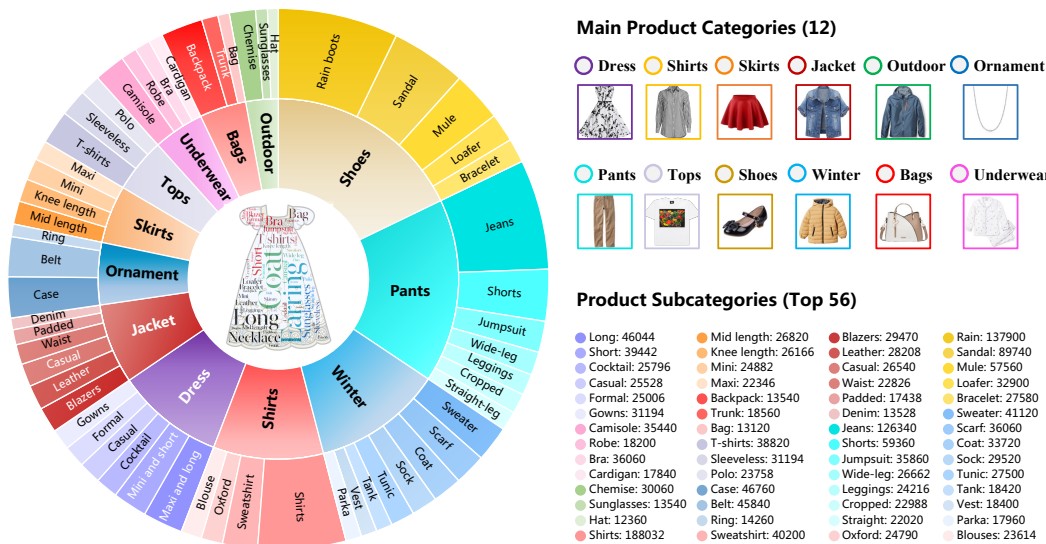

Figure 2: Top categories and distribution in our proposed FashionMT dataset. We have listed 12 main product categories and the top 56 product subcategories to provide a clearer presentation.

Table 1: Comparison with other MTCIR datasets. *MT* stands for Multi-turn. In a *transaction*, there are multiple *turns*. *Length* denotes the modified text's average length. *Categories* denotes the number of finest subcategories, while *Product type* lists the typical product categories.

| Datasets | # Images | # Transactions | # Turns | # Categories | Length | Product type |
|---|---|---|---|---|---|---|
| MT FashionIQ Yuan & Lam | 74,381 | 11,505 | 26,506 | 3 | 10.7 | shirt, top-tee, dress |
| MT Shoes Pal et al. | 15,661 | 4,097 | 11,346 | 10 | 5.2 | boots, sneakers, clogs, etc. |
| **FashionMT (ours))** | 1,067,688 | 247,911 | 743,733 | 95 | 24.3 | shirt, top-tee, dress, shoes, coat, pants, bag, ornament, etc. |

by capturing the distinctions between reference and target image pairs. The framework consists of the following steps: (i) Image Selection: Selecting $N + 1$ images from a product subcategory for $N$ turns in a transaction. (ii) Caption Generation: Generating captions for these images using an image captioning model. (iii) Base Modification Generation: Employing a large language model (LLM) to describe the differences between image captions from adjacent turns. (iv) Retrospective Modification Generation: Determining the specific turns requiring retrospective analysis and generating corresponding modified text based on the intersection of attributes between the most recent image and images from previous turns.

Specifically, we generate the captions using the prompt: "*Question: Describe the product. Answer:*". For generating base modified text, the prompt is: "*The reference depicts {REF}, and the target depicts {TAR}. Describe the modifications to transform the reference into the target*", where *REF* and *TAR* represent the captions of the reference and target images within a single turn, respectively.

To better align with retrospective needs in real-world scenarios, we have established two scenarios for generating retrospective-based modified text: **rollback** and **combination**. In the rollback setting, similar to base transaction generation, modifications are generated between a specified reference and the target by rolling back. An example under this setup is: "*Compared to the most recent turn, I still prefer the item from the second turn. Building on that, I like...*". In the combination setting, users combine attributes from multiple images in historical turns to formulate modification requests. An example under this setup would be: "*I like ... from the first turn, and ... from the second turn*". In this setup, the modified text consists of two parts: the initial segment encompasses common attributes earmarked for retention, prefaced by the prompt "*Keep the {Attr} in the {ID} turn*" where *Attr* represents common properties like color, logo, pattern, etc., and *ID* signifies the turns sharing commonalities with the target. Meanwhile, the subsequent segment delineates additional modifica-

tion requisites, prefaced by the prompt "*the reference images depict REFs, the target depicts TAR. Describe the distinctiveness of the target:*".

## 3.2 DATASET STATISTICS

The data distribution of FashionMT is illustrated in Figure 2. Detailed information and a comparison with existing datasets, MT FashionIQ Yuan & Lam (2021) and MT Shoes Pal et al. (2023), are presented in Table 1. FashionMT significantly surpasses existing datasets in both scale and richness, featuring 14 times more images than MT FashionIQ and nearly 10 times more categories than MT Shoes. By leveraging the Modification Generation Framework, FashionMT enables the efficient construction of high-quality transactions, resulting in a dataset that is 27 times larger than MT FashionIQ. Additionally, FashionMT provides more detailed modified text, with an average length twice that of MT FashionIQ. As a dataset tailored specifically for MTCIR task, FashionMT offers more comprehensive and realistic data support. For more details on our proposed dataset, including setup explanations and quality control, please refer to Section 7.3.

## 4 APPROACH

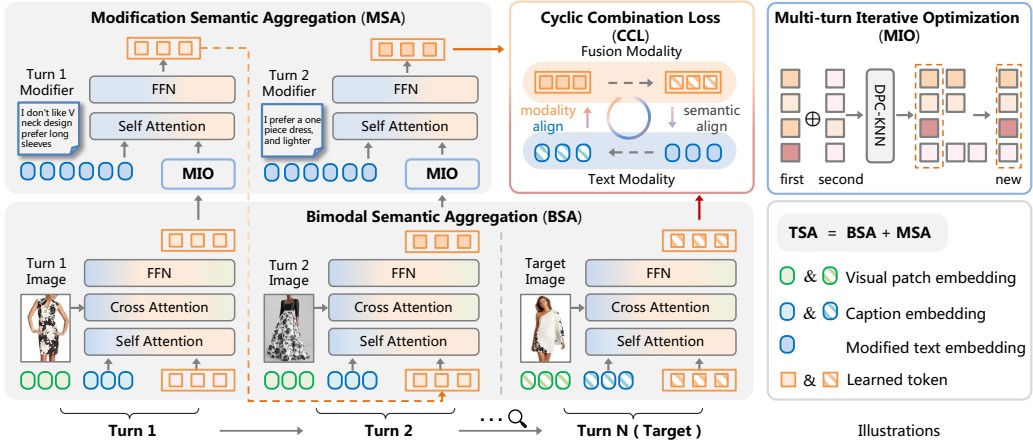

Figure 3: The architecture of the MAI model. For each turn, images, captions, and modified text are progressively aggregated through BSA and MSA, with MIO preserving core information across turns, and CCL constraining the training process. For simplicity, we illustrate two retrieval turns.

### 4.1 PROBLEM FORMULATION

In our task setup, we provide the previous $N-1$ turns' multimodal data, which consists of predefined reference images with captions and modified text, aiming to retrieve the most suitable target image based on the final modified text. We represent the image patch embedding, caption embedding, and modified text embedding of the $n$-th turn as $v_n \in \mathcal{V}$, $c_n \in \mathcal{C}$, and $m_n \in \mathcal{M}$ respectively. Furthermore, we distinguish among the reference image embedding, reference caption embedding, target image embedding, and target caption embedding as $v_n^r$, $c_n^r$, $v_n^t$, and $c_n^t$ respectively.

### 4.2 MULTI-TURN AGGREGATION-ITERATION (MAI) MODEL

The architecture of MAI is depicted in Figure 3. We will introduce Bimodal Semantic Aggregation (BSA) and Modification Semantic Aggregation (MSA), which are part of the Two-stage Semantic Aggregation (TSA), along with the Multi-turn Iterative Optimization (MIO).

**Bimodal Semantic Aggregation (BSA).** In the $n$-th turn, we first conduct a lexical analysis on the modified text to determine if there is a rollback operation. We established a template for automatically generating modified text with Rollback instructions to facilitate benchmark construction.

The template includes phrases such as: "Compared to this one I prefer the {}, and", "I would rather choose the {}, and", where {} denotes rollback turn descriptions, such as "Turn 2: White off-shoulder lace short sleeve." Rollback operations are executed when modified text match the template. If so, the reference image is designated as the image from the specified rollback turn. If not, the default reference image for the $n$-th turn is adopted. We extract visual patch embeddings $v_n$ of images using a frozen visual encoder. The effectiveness of the Q-Former architecture Li et al. (2023) in integrating vision-text embeddings has been validated in prior studies Bai et al. (2024); Hu et al. (2024). Hence, our BSA transfers this framework to adapt to the MTCIR task. Through learned tokens $t_n$, BSA initially learns the bimodal semantics of images and their corresponding captions before interacting with modified text. This strategy employs captions as a transition, enhancing modality relevance during interaction with modified text, as elaborated in Section 4.3. A fixed text encoder extracts caption embeddings $c_n$, interacting with learned tokens $t_n$ in BSA's self-attention layers. In the cross-attention layers, they engage with visual patch embeddings $v_n$. After BSA, learned tokens aggregate multimodal semantics from reference images and captions, denoted as $t_n^{r,\text{BSA}}$. As the target side lacks modified text, this embedding is directly used for training loss constraints and inference distance measurement.

**Multi-turn Iterative Optimization (MIO).** Despite learned tokens being more space-efficient than visual embeddings Li et al. (2023), storing these tokens for each turn still results in significant space consumption. Additionally, fashion images encompass various attributes such as color, style, size, etc Tian et al. (2023); Chen et al. (2023b); Han et al. (2023). Multi-turn retrieval often revolves around the same subcategory of products, resulting in similar attributes across the images involved in multiple turns. Therefore, we propose a parameter-free mechanism to optimize and retain the key attributes throughout multi-turn interactions.

Specifically, we concatenate $t_{n-1}^{r,\text{MIO}}$ from the previous turn with $t_n^{r,\text{BSA}}$ from the current turn to obtain $t_n^{r,\text{MIO}}$. Our objective is to preserve key semantic tokens while discarding redundant ones from the learned tokens $t_n^c$. This process involves several steps. (i) Clustering. We apply an efficient density peaks clustering based on $k$-nearest neighbors (DPC-kNN) algorithm Du et al. (2016). The learned tokens $t_n^c$ are clustered into $k$ groups and the clustering operation is formulated as follows:

$$\text{cluster}(t_n^c, k) = \arg\min_C \sum_{i=1}^{k} \sum_{v \in C_i} ||v - c_i||^2 \tag{1}$$

where $C$ represents the clusters, $C_i$ represents the $i$-th cluster, and $c_i$ represents the centroid of the $i$-th cluster. (ii) Density Estimation. After clustering, the density of each cluster is estimated based on the distances between the tokens within the cluster and learned tokens with lower densities are filtered out to enhance clustering efficiency. The density estimation is calculated as follows:

$$\text{density}(v) = \exp(-\frac{1}{k} \sum_{u \in \text{Nei}(v)} ||v - u||^2) \tag{2}$$

where $\text{Nei}(v)$ represents the neighboring tokens of $v$. (iii) Pruning. Tokens with low density are eliminated to ensure that only the most semantically significant tokens are retained. To achieve this, each token is assigned a score, computed as the product of its density and its distance to the nearest neighbor. The top $k$ tokens with the highest scores are then selected as the optimized tokens.

$$\text{score}(v) = \text{density}(v) \times \text{dist}(v) \tag{3}$$

where $\text{dist}(v)$ represents the distance of token $v$ to its nearest neighbor. The final tokens, denoted as $t_n^{r,\text{MIO}}$, are obtained by selecting the tokens with the top-$k$ scores. Through the optimization process described above, MIO effectively preserves learned tokens carrying discriminative semantics while discarding tokens with relatively less semantic importance, thereby saving computational resources.

**Modification Semantic Aggregation (MSA).** During the MSA stage, we engage the tokens $t_n^{r,\text{MIO}}$, which encapsulate reference semantics, with the modified text embedding $m_n$. By employing a frozen text encoder to extract embeddings $m_n$, we concatenate them with learned tokens $t_n^{r,\text{MIO}}$ before feeding them into the self-attention layer. Subsequently, we employ a linear and normalization layer on the learned tokens to map them, producing a reference-side embedding $t_n^r$. This embedding concurrently embodies multimodal semantics from the reference and modified text.

It is important to note that in the combination setting, due to the involvement of multiple historical images, BSA aggregates the bimodal embeddings by concatenating the learned tokens from previous turns with their corresponding image captions. Subsequently, these embeddings are semantically aggregated with the modified text in the MSA.

## 4.3 THEORETICAL INSIGHTS

In this section, we justify the rationale behind introducing captions for bimodal semantic aggregation and explain how our approach outperforms a naive solution. In the MTCIR task, we hypothesize that the transition from the initial image to the final target image occurs by gradually introducing specific attributes related to the modified text in each turn, denoted as $v_n^t - v_n^r \sim \mathcal{N}(m_n, \frac{1}{N}I)$. Ideally, the visual increments $v_n^t - v_n^r$ should correspond to the textual modifications $m_n$, i.e., $v_n^r + m_n = v_n^t$, which can be supervised using the following similarity loss:

$$\mathcal{L}_{\text{sim}} = \frac{1}{B} \sum_{i=1}^{B} \left( 1 - \frac{|v_n^{ri}| + |m_n^i|}{2} \cdot |v_n^{ti}| \right) \tag{4}$$

where for simplicity, given an embedding $x$, $|x|$ stands for the normalized form $\frac{x}{||x||}$.

However, the effectiveness of the aforementioned supervision is constrained by: (i) the inherent modality gap between texts and images; (ii) the semantic disparity between the additional textual attributes and visual items. To mitigate these gaps, we propose leveraging image captions to (i) align with the modality of modified texts; (ii) match the semantics of visual items. We can adopt a naive method Huang et al. (2023), replacing the visual embeddings in Eq. 4 with the corresponding caption text embeddings for cross-modal constraints. This results in the following **naive** cross-modal loss by replacing images with their corresponding captions:

$$\mathcal{L}_{\text{naive}} = \mathcal{L}_{\text{sim}} + \frac{1}{B} \sum_{i=1}^{B} \left[ 1 - \frac{1}{2} \left( \frac{|v_n^{ri}| + |m_n^i|}{2} \cdot |c_n^{ti}| + \frac{|c_n^{ri}| + |m_n^i|}{2} \cdot |v_n^{ti}| \right) \right] \tag{5}$$

Although $\mathcal{L}_{\text{naive}}$ aligns visual increments with textual modifications, bridging both modality and semantic gaps in the meantime, the separate optimization in modality and semantic space may affect each other in the training process. To make further efforts, we propose that the reference and target images should undergo **pre**-fusion with caption text to achieve an intermediate state that is closer in modality and semantics to the modified text. The paradigm of this process is represented as follows:

$$\mathcal{L}_{\text{pre}} = \mathcal{L}_{\text{sim}} + \frac{1}{B} \sum_{i=1}^{B} \left( 1 - \frac{(|v_n^{ri}| + |m_n^i|) + (|c_n^{ri}| + |m_n^i|)}{4} \cdot \frac{|v_n^{ti}| + |c_n^{ti}|}{2} \right) \tag{6}$$

Furthermore, we give theoretical justifications for the effectiveness of the proposed pre-fusion loss:

**Proposition 1**: *Let $\mathcal{O}(GError(\mathcal{L}_{\text{pre}}))$ and $\mathcal{O}(GError(\mathcal{L}_{\text{naive}}))$ be the upper bound of generalization error of the above two losses. Then for any hypothesis $\mathcal{L}_{\text{pre}}, \mathcal{L}_{\text{naive}}$ in $\mathcal{H} : \mathcal{V} \times \mathcal{C} \times \mathcal{M} \to [0, 1]$ and $1 > \delta > 0$, it holds that:*

$$\mathcal{O}(\text{GError}(\mathcal{L}_{\text{pre}})) \leq \mathcal{O}(\text{GError}(\mathcal{L}_{\text{naive}})) \tag{7}$$

*with probability at least $1 - \delta$, given that the visual and textual encoders are reliable for generating positively correlated embeddings in the $n$-th turn and clustering embeddings with the same modality.*

## 4.4 OPTIMIZATION AND INFERENCE

**Training.** Given the guiding role of modified text in retrieval Chen et al. (2024a), we design the **Cyclic Combination Loss (CCL)** to align semantically similar fused modality with text modality, thereby preserving the critical semantics within the textual modality. Specifically, we employ a batch-based classification loss commonly used in CIR and MTCIR tasks Pal et al. (2023); Wen et al. (2024); Chen et al. (2024a), which is defined as:

$$\mathcal{L}_{\text{B}}(r_q, r_t) = \frac{1}{B} \sum_{i=1}^{B} -\log \frac{\exp \kappa(r_q^i, r_t^i)}{\sum_{j=1}^{B} \exp \kappa(r_q^i, r_t^j)} \tag{8}$$

where $B$ represents the batch size, the kernel $\kappa()$ is the inner product resulting in cosine similarity. $r_q$ denotes the reference-side representation, and $r_t$ signifies the target-side representation.

Inspired by $\mathcal{L}_{\text{pre}}$'s paradigm in Section 4.3 and 7.1, our loss function incorporates three constraints on embeddings after Bimodal Semantic Aggregation pre-fusion, along with an additional constraint on the text modality. For the $n$-th turn, the cyclic constraints involve the following four sets of embeddings: learned tokens $t_n^r$ from MSA, encompassing semantics of the reference image, caption, and modified text; learned tokens $t_n^{tg}$ from BSA, containing semantics of the target image and its caption; the modified text embedding $m_n$ and the caption text feature of the target image $c_n^{tg}$.

In line with previous works in the MTCIR task, our overall Cyclic Combination Loss $\mathcal{L}_{\text{CCL}}$ for $N$ turns is composed of the cumulative losses from each turn:

$$\mathcal{L}_{\text{CCL}} = \sum_{n=1}^{N} \mathcal{L}_{\text{B}}(t_n^r, t_n^{tg}) + \mathcal{L}_{\text{B}}(t_n^{tg}, m_n) + \mathcal{L}_{\text{B}}(m_n, c_n^{tg}) + \mathcal{L}_{\text{B}}(c_n^{tg}, t_n^r) \tag{9}$$

**Inference.** At the conclusive $N$-th turn, $t_{N-1}^{r,\text{MIO}}$ encompasses key multimodal semantics from prior turns. Upon interacting with the modified text through MSA, we derive the reference-side embedding $t_N^r$. Meanwhile, on the gallery side, the bimodal embedding of the image and its caption, $t_N^{tg}$, is computed. Retrieval matching ensues by evaluating the similarity between $t_N^r$ and $t_N^{tg}$.

## 5 EXPERIMENT

### 5.1 EXPERIMENTAL SETTING

**Implementation Details.** We adopt BLIP-2 Li et al. (2023) with the Flan-t5-xxl language model Chung et al. (2024) for image captioning and Xwin-13B-V0.2 Ni et al. (2024) as the LLM. Optimization is performed using AdamW Loshchilov & Hutter (2019) with a batch size of 16, an initial learning rate of 1e-5, and cosine annealing. Training runs for 50 epochs, while inference uses a batch size of 2048. All model training and inference are conducted on 8 V100 GPUs. The number of learned tokens is fixed at 32, and 32 tokens are retained each turn through the MIO. Q-Former parameters are initialized with $\text{blip2\_pretrain\_vitL}$, consistent with SPRC Bai et al. (2024).

**Representative Methods.** We select representative methods from five different categories for comprehensive performance comparison: **(i) MTCIR** methods including FashionNTM Pal et al. (2023) and CFIR Yuan & Lam (2021). **(ii) STFIR+NTM**: DQU-CIR Wen et al. (2024), single-turn methods FashionERN Chen et al. (2024a), and SPRC Bai et al. (2024), integrated with the multi-turn method FashionNTM. **(iii) ZS-CIR**: Pic2word Saito et al. (2023), Context-I2W Tang et al. (2024), and Image2Sentence Du et al. (2024), also integrated with FashionNTM. Additionally, since LLMs inherently support multi-turn interactions, we select several MLLMs as stronger baselines for comparison and fine-tune them on FashionMT using their original training methods. The selected baselines include: **(iv) Retrieval-capable MLLMs:** Fromage Koh et al. (2023), GILL Koh et al. (2024). We use the [IMG] and [RET] tags provided by these methods for retrieval. **(v) Interleaved MLLM:** MLLMs designed for interleaved multiple images and text, including MMICL Zhao et al. (2024a) and Flamingo Alayrac et al. (2022)-9B. For these methods, we perform retrieval by encoding the target's description text with the final-round LLM. All methods use ViT-L Radford et al. (2021) as the visual backbone for fair comparison.

**Evaluation Metrics.** Consistent with existing multimodal retrieval tasks Pal et al. (2023); Wen et al. (2024), we use the standard top-K recall metric to evaluate models' performance, denoted as R@K. Specifically, we adopt R@1, R@5, R@10, R@20 and their mean as the evaluation metrics.

### 5.2 RESULTS

**Quantitative Analysis.** Experimental results on FashionMT are shown in Table 2. Benefiting from the strong multimodal fusion capability of the BLIP-2 architecture, methods such SPRC Bai et al. (2024) demonstrate performance advantages. Building upon this, our TSA and CCL incorporate captions as a transition, leveraging their semantic alignment with references and consistency with

modified text. Furthermore, the proposed MIO effectively retains key semantics across multiple turns. Consequently, MAI significantly outperforms existing methods, achieving a remarkable 8.63 improvement in the Mean metric over the SOTA method SPRC.

Table 2: Results on our proposed FashionMT dataset.

| Method | Combination | | | | Rollback | | | | Mean |
|--------|------|------|------|------|------|------|------|------|------|
| | R@1 | R@5 | R@10 | R@20 | R@1 | R@5 | R@10 | R@20 | |
| CFIR (SIGIR'21) | 11.70 | 23.09 | 30.89 | 40.14 | 8.25 | 22.63 | 31.04 | 41.79 | 26.19 |
| Pic2word (CVPR'23) | 13.35 | 27.12 | 35.42 | 45.40 | 8.69 | 23.98 | 33.15 | 44.41 | 28.94 |
| GILL (NeurIPS'23) | 19.54 | 38.17 | 47.63 | 56.14 | 9.12 | 24.56 | 33.62 | 42.77 | 33.95 |
| Fromage (ICML'23) | 19.45 | 39.12 | 49.00 | 59.65 | 10.12 | 26.54 | 34.97 | 45.61 | 35.56 |
| FashionNTM (ICCV'23) | 18.98 | 38.51 | 48.35 | 58.30 | 10.73 | 27.71 | 37.66 | 49.85 | 36.26 |
| FashionERN (AAAI'24) | 20.36 | 41.37 | 50.18 | 60.51 | 11.42 | 29.67 | 41.02 | 52.98 | 38.44 |
| Flamingo (NeurIPS'22) | 21.38 | 44.17 | 55.16 | 63.09 | 11.55 | 28.18 | 37.81 | 48.76 | 38.76 |
| DQU-CIR (SIGIR'24) | 20.57 | 42.32 | 52.33 | 62.03 | 12.59 | 31.69 | 42.79 | 54.68 | 39.88 |
| Context-I2W (AAAI'24) | 30.62 | 51.84 | 62.50 | 71.75 | 12.63 | 32.98 | 45.48 | 59.30 | 45.89 |
| Image2Sentence (ICLR'24) | 32.44 | 53.71 | 65.16 | 74.52 | 15.79 | 36.87 | 50.17 | 64.56 | 49.15 |
| MMICL (ICLR'24) | 39.17 | 60.89 | 70.28 | 79.89 | 18.46 | 43.53 | 57.05 | 69.66 | 54.87 |
| SPRC (ICLR'24) | 39.28 | 62.42 | 72.11 | 80.23 | 23.31 | 49.79 | 62.11 | 74.82 | 58.01 |
| **MAI (ours)** | **51.51** | **74.67** | **80.66** | **86.52** | **28.94** | **58.89** | **70.42** | **81.50** | **66.64** |

**Qualitative Analyses.** In Figure 4, we compare MAI with two representative methods, Fashion-NTM and SPRC. MAI effectively handles fine-grained demands by leveraging TSA and CCL for efficient aggregation of image-caption semantics, making it responsive to domain-specific terms like "crepe fabric" and "vintage design." Furthermore, MAI addresses retrospective-based needs by utilizing the MIO component to retain multi-turn historical key information, enabling precise interpretation of vague expressions such as "strap design."

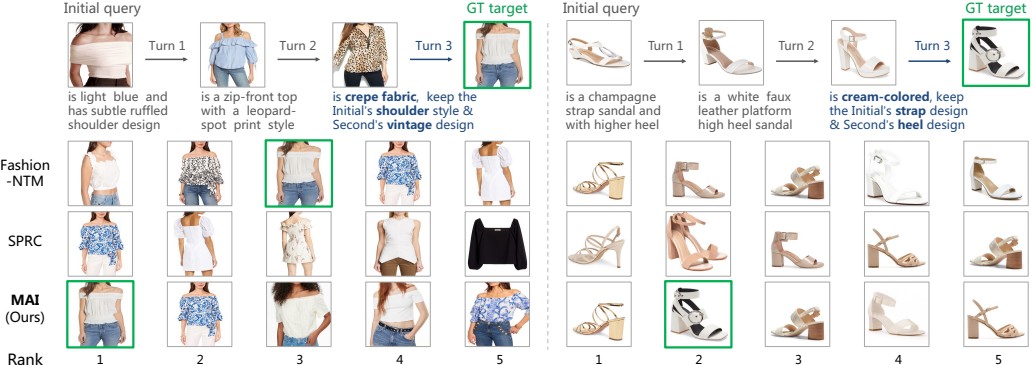

Figure 4: Qualitative results for the last turn in the FashionMT dataset. The top 5 retrieval results of MAI compared with two representative methods are shown.

Table 3: Ablation study on different components of the MAI model. Mean-Combination and Mean-Rollback denote the mean recall under the combination and rollback settings.

| Settings | Mean-Combination | Mean-Rollback | Mean |
|----------|------------------|---------------|------|
| Base | 58.69 | 41.49 | 50.04 |
| Base + TSA | 69.22 | 55.74 | 62.48 |
| Base + MIO | 64.17 | 47.03 | 55.60 |
| Base + TSA + CCL | 72.31 | 58.83 | 65.57 |
| Base + TSA + MIO | 71.19 | 58.19 | 64.69 |
| Base + TSA + CCL + MIO (**MAI**) | **73.34** | **59.94** | **66.64** |

## 5.3 ABLATION STUDIES

**Effects of Different Components.**    Our baseline method employs Q-Former from BLIP-2 Li et al. (2023) for reference and modified text semantic fusion and adopts the multi-turn information aggregation model from FashionNTM for task adaptation. We gradually add the TSA, CCL and MIO, comparing their performance in both combination and rollback settings. Table 3 demonstrates the positive contribution of each component to performance improvement in both settings.

Table 4: Effects of TSA and CCL. Mean-C and Mean-S denote using caption adaptation and single-turn results.

| Method | Mean-C | Mean-S |
|---|---|---|
| FashionERN 2024a | 44.47 | 50.29 |
| Image2Sentence 2024 | 47.61 | 51.64 |
| FashionNTM 2023 | 48.03 | 45.75 |
| MMICL 2024a | 58.67 | 47.28 |
| SPRC 2024 | 62.90 | 52.66 |
| **TSA + CCL** | **65.57** | **53.73** |

Table 5: The comparison between MIO and other methods on memory cost and average retrieval metrics. "$N$" represents # turns.

| Method | Memory Cost (MB) | Mean |
|---|---|---|
| None | 0 | 50.04 |
| Concat | $64 \times N$ | 50.56 |
| LSTM 2012 | $957 + 64 \times N$ | 52.08 |
| GRU 2017 | $858 + 64 \times N$ | 51.80 |
| NTM 2023 | $1270 + 64 \times N$ | 53.19 |
| **MIO** | 64 | **55.60** |

Table 6: Effects of each loss in CCL. For simplicity, we denote $\mathcal{L}_B(x, y)$ as $\mathcal{L}(x, y)$.

| Settings | w/ CCL (total) | w/o CCL | w/o $\mathcal{L}(t_n^r, t_n^{tg})$ | w/o $\mathcal{L}(t_n^{tg}, m_n)$ | w/o $\mathcal{L}(m_n, c_n^{tg})$ | w/o $\mathcal{L}(c_n^{tg}, t_n^r)$ |
|---|---|---|---|---|---|---|
| Recall | **65.57** | 62.48 | 63.54 | 64.39 | 64.51 | 64.90 |
| $\Delta$ | - | -3.09 | -2.03 | -1.18 | -1.06 | -0.67 |

**Effects of TSA and CCL.**    We further conduct two sets of experiments, as shown in Table 4 with Mean-C and Mean-S. (i) Caption adaptation. We adapt several representative methods to a two-stage fusion process, allowing the reference image to interact with both the caption text and the modified text. Specifically, FashionERN, Image2Sentence, and FashionNTM utilize the Combiner Baldrati et al. (2022) widely employed in this field, for interaction with caption embeddings. (ii) Single-turn retrieval. We evaluate performance using the first turn from FashionMT. The results in Table 4 indicate that the two-stage fusion significantly improves the performance of the methods. Additionally, the combination of TSA and CCL effectively integrates the critical semantics from the modified text. Consequently, it achieves superior retrieval performance compared to existing methods in both settings, shown in Table 4. We also conduct ablation experiments for each loss in CCL. Since CCL computes losses based on the outputs from TSA, it requires TSA to be present, resulting in 6 settings. Results in Table 6 show that each loss contributes to performance gains.

For further ablation studies on performance in existing datasets, reducing modality gap, and rollback setting, please refer to Section 7.1.

## 6 CONCLUSION AND DISCUSSION

In this paper, we have constructed the first dataset specifically designed for Multi-turn Composed Image Retrieval, named FashionMT. We also propose MAI model, a multi-turn key information-aware approach that uses paired captions as a transition for better semantic consistency and modality alignment while adaptively filtering and preserving significant attributes to reduce spatial occupancy. We have conducted extensive experiments on FashionMT and observed that MAI achieves state-of-the-art performance, demonstrating its usefulness and effectiveness.

**Limitations.**    As the first dedicated MTCIR dataset, we standardize the number of turns to 3 for method comparison. However, real-world scenarios may involve more diverse transactions and cover more general contexts beyond e-commerce, aligning with our ongoing development efforts. Furthermore, we aim to upgrade our model with integrated dialogue and retrieval capabilities.

## ACKNOWLEDGMENTS

This work was supported by the grants from the National Natural Science Foundation of China (61925201, 62132001, 62432001, 62373043) and Beijing Natural Science Foundation (L247006, 4252020).

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

# 7 APPENDIX

## 7.1 PROOF OF PROPOSITION 1

For simplicity, given two representations $x, y$, we denote their similarity score as $s(x, y) = x \cdot y$. We first compare the $i$-th similarity score terms in $L_{\text{sim}}$ and $L_{\text{pre}}$ respectively:

$$S_{\text{naive}}^i = \frac{1}{2}\left(\frac{|v_n^{ri}| + |m_n^{ri}|}{2} \cdot |c_n^{ti}| + \frac{|c_n^{ri}| + |m_n^{ri}|}{2} \cdot |v_n^{ti}|\right)$$

$$= \frac{1}{4}[s(|v_n^{ri}| + |m_n^{ri}|, |c_n^{ti}|) + s(|c_n^{ri}| + |m_n^{ri}|, |v_n^{ti}|)],$$

$$S_{\text{pre}}^i - S_{\text{naive}}^i = \frac{1}{8}[s(|v_n^{ri}|, |v_n^{ti}|) + s(|c_n^{ri}|, |c_n^{ti}|) - s(|v_n^{ri}|, |c_n^{ti}|) - s(|c_n^{ri}|, |v_n^{ti}|)].$$

The modality gaps between visual images and textual captions indicate that representations within the same modality are closer to each other, leading to higher similarity scores, i.e., $s(|v_n^{ri}|, |v_n^{ti}|) > s(|v_n^{ri}|, |c_n^{ti}|)$, and $s(|c_n^{ri}|, |c_n^{ti}|) > s(|c_n^{ri}|, |v_n^{ti}|)$. Therefore, $S_{\text{pre}}^i - S_{\text{naive}}^i > 0$. Notice that:

$$L_{\text{naive}} = L_{\text{sim}} + \frac{1}{B}\sum_{i=1}^{B}(1 - S_{\text{naive}}^i),$$

$$L_{\text{pre}} = L_{\text{sim}} + \frac{1}{B}\sum_{i=1}^{B}(1 - S_{\text{pre}}^i),$$

where $L_{\text{naive}} > L_{\text{pre}}$. Furthermore, based on the Rademacher Complexity Theory Mohri (2018), the upper bound of generalization errors is estimated as follows, with probability at least $1 - \delta$:

$$E[L_{naive}] \leq E[L_{\text{sim}}] + \frac{1}{B}\sum_{i=1}^{B}(1 - S_{\text{naive}}^i) + 2R_B(G) + \sqrt{\frac{\log\frac{1}{\delta}}{2B}}$$

$$:= O(\text{GError}(\text{L}_{\text{naive}})),$$

$$E[L_{\text{pre}}] \leq E[L_{\text{sim}}] + \frac{1}{B}\sum_{i=1}^{B}(1 - S_{\text{pre}}^i) + 2R_B(G) + \sqrt{\frac{\log\frac{1}{\delta}}{2B}}$$

$$:= O(\text{GError}(\text{L}_{\text{pre}})),$$

where $R_B(G)$ is the Rademacher Complexity of the family of all possible loss functions, independent of our design for loss functions. From the above analysis, we have $O(\text{GError}(L_{\text{pre}})) < O(\text{GError}(L_{\text{naive}}))$, which indicates the superiority of our pre loss $L_{\text{pre}}$ to the original naive cross-modal loss $L_{\text{naive}}$. □

### 7.2 MORE ABLATION STUDIES

**Validation on Existing Datasets.** Despite the limitations of existing datasets, we further validate the effectiveness and generalization of our approach by adding performance comparisons on the real datasets MT FashionIQ Yuan & Lam (2021) and MT Shoes Pal et al. (2023). The results from Table 7 shows that our proposed approach, MAI, achieves the best performance in all settings due to its fine-grained semantic capture and efficient modality alignment.

Table 7: Validation on existing datasets MT FashionIQ and MT Shoes.

| Method | Train on FashionMT Test on MT FashionIQ | Train on FashionMT Test on MT Shoes | Train on FashionMT Test on FashionMT | Means |
|---|---|---|---|---|
| FashionNTM Pal et al. | 42.3 | 25.6 | 36.26 | 34.72 |
| Image2Sentence Du et al. | 43.8 | 28.2 | 49.15 | 40.38 |
| MMICL Zhao et al. | 46.9 | 29.8 | 54.87 | 43.86 |
| SPRC Bai et al. | 48.0 | 28.2 | 58.01 | 44.74 |
| **MAI (ours)** | **50.6** | **33.8** | **66.64** | **50.35** |

**Modality Gap.** We observe that recent works enhance feature-level representations to reduce modality gaps. We compare our approach with these methods, shown in the Table 8. The results show that noise addition methods are effective for large modality gaps, but once our approach reduces the gap through aligning modalities and semantics, the gains are limited.

Table 8: Comparison of various methods for reducing modality gaps.

| Settings | Recall | Settings | Recall |
|---|---|---|---|
| Base | 50.04 | Base | 50.04 |
| + Mixing Vouitsis et al. | 49.87 | **+ ours** | 65.57 |
| $+ N(0,1)$ | 52.17 | + ours $+ N(0,1)$ | 65.23 |
| $+ U(-1,1)$ | 52.09 | + ours $+ U(-1,1)$ | 65.30 |
| $+ N(0,1) \times U(-1,1)$ Gu et al. | 54.88 | + ours $+ N(0,1) \times U(-1,1)$ | 65.66 |

**Ablation Study on Rollback.** In the current setup for handling Rollback operations, the reference image for the current turn is replaced with the specified rollback image. We conduct comparative experiments under various Rollback settings: (1) Replace: the current setting. (2) Ignore: no replacement is performed. (3) Random: selecting randomly from previous turns. (4) Blend: using the PIL library's Image.blend() to merge two images into one. The results from Table 9 indicate that since the Rollback operation approximates redefining the current local optimal point, the setting Replace achieves the best performance.

Table 9: Ablation study on Rollback setting

| Settings | Replace | Ignore | Random | Blend |
|---|---|---|---|---|
| Recall | **59.94** | 49.59 | 53.62 | 57.80 |

## 7.3 Further clarification on the FashionMT dataset

We further clarify the setting, utility, quality control, and benefits of our FashionMT dataset below.

**Explanation of Modified Text in Multi-turn.** Our current approach for constructing modified text is based on two main reasons:

- **User Target Ambiguity.** Humans often make decisions heuristically, so selecting while browsing aligns with human intuition Todd & Gigerenzer (2000); Schubert (2023). In the e-commerce domain, our analysis of multi-turn interaction data from a well-known platform shows that users frequently experience "**target ambiguity**" during online shopping. Initially, users are unsure of the exact target and its details, and they need to select and refine attributes throughout the multi-turn interaction process. This behavior is also supported by psychological studies Yoo & Sarin (2018); Sung et al. (2023). To simulate this, we use combination and rollback settings to better mirror real-world scenarios.

- **Benchmark for Multi-turn.** Initially, we explored constructing the dataset by describing the difference between the current and target images. However, this "clear goal" setting resulted in models achieving precise retrieval within 1-2 turns. This results in the multi-turn retrieval task **degrading into a single-turn** retrieval task, thus failing to serve as a benchmark requiring algorithms to integrate information from multiple historical interactions.

**Utility.** Although the FashionMT is synthetically constructed, we conduct an in-depth analysis of user behaviors during multi-turn purchases on a famous e-commerce platform. We categorize these behaviors into two representative scenarios: "combination" and "rollback", aiming to replicate the real-world process where users refine their choices through iterative comparisons. Compared to existing datasets that concatenate single-turn retrieval data, FashionMT more accurately reflects real-world scenarios.

Table 10: Quality assessments among multi-turn datasets.

| Datasets | Acc | HA | Gra | Con | Cov | Mean |
|---|---|---|---|---|---|---|
| MT FashionIQ Yuan & Lam | 93.3 | 43.1 | 67.2 | 86.4 | 70.2 | 72.0 |
| MT Shoes Pal et al. | 95.1 | 65.6 | 76.3 | 91.3 | 80.9 | 81.8 |
| **MAI (ours)** | 96.2 | 98.7 | 91.3 | 90.5 | 93.2 | 94.0 |

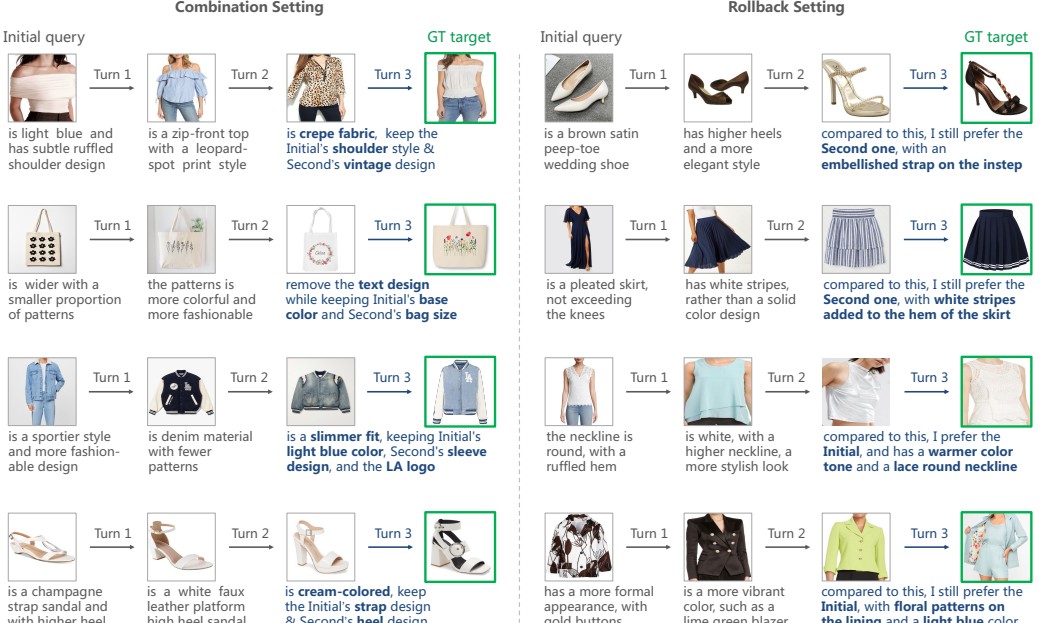

Figure 5: Examples of image sequences across multiple turns in the Combination and Rollback settings from our proposed FashionMT dataset.

**Quality Control.** Additionally, to validate its utility, we conduct a quality assessment of FashionMT and existing datasets, scoring them on a scale from 1 to 5 in the following aspects:

- Accuracy (**Acc**): Whether the modified text reflects the actual differences between images, with 1 being very inaccurate and 5 being very accurate.
- Historical Awareness (**HA**): Whether the modified text involves attributes from previous turns, with 1 being not involved and 5 being fully involved.
- Granularity (**Gra**): Whether the text provides enough detail to cover subtle differences between images, with 1 being lacking detail and 5 being very detailed.
- Consistency (**Con**): Whether the differences between items in multi-turn retrieval are realistic, with 1 being unrealistic and 5 being very realistic.
- Coverage (**Cov**): Whether the description covers all major differences between items, with 1 being minimal coverage and 5 being comprehensive coverage.

We provided explanations of the rating requirements to 20 e-commerce platform staff members and calculated the average scores independently. The scores were then converted into percentages, as shown below. Specifically, we randomly selected 20% of the data from each dataset for manual scoring without informing the evaluators of the data source. Quality assessments among multi-turn datasets are shown in Table 10.

## 7.4 MORE VISUALIZATION

**Dataset Examples.** To facilitate a better understanding of our newly proposed dataset, FashionMT, we present data samples under the Combination and Rollback settings in Figure 5. In each

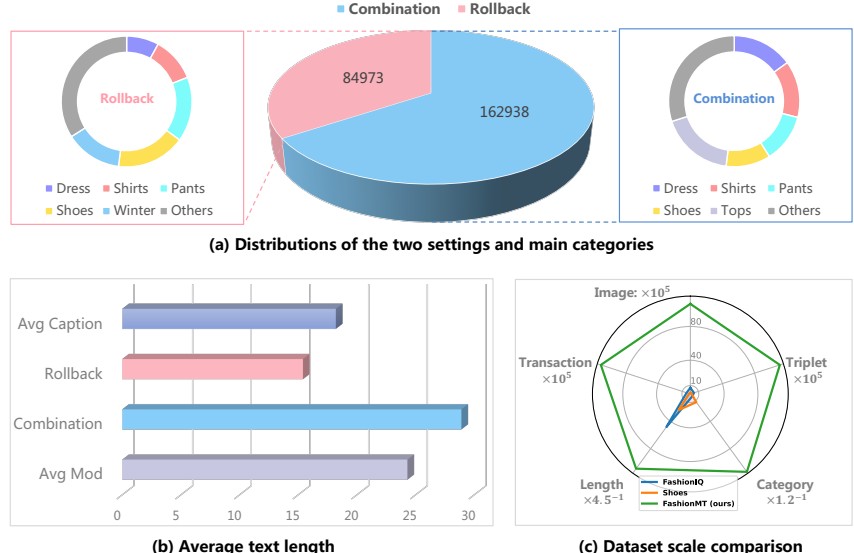

Figure 6: (1) Proportions of Combination and Rollback settings and main category distributions; (2) Average lengths of Modified text and captions; (3) Scale comparison of FashionMT with existing multi-turn datasets.

transaction, the Ground Truth is highlighted with a green bounding box, and the retrospective-based modified text is marked in dark blue.

**Dataset Statistics Visualizations.** Figure 6 provides visualizations of various statistics in the FashionMT dataset. (1) The proportions of Combination and Rollback settings and their respective main categories. (2) The average length of modified text, along with separate averages for Combination and Rollback settings, and the average caption length. (3) A scale comparison between FashionMT and existing multi-turn datasets, MT FashionIQ and MT Shoes.

**Modality gap.** As shown in Figure 7, we visualize the modality gap between the query and target sides in the final round using t-SNE on FashionMT and existing datasets. Leveraging captions as a bridge between visual and textual modalities, our proposed MAI approach effectively reduces the modality gap between the query and target sides.

**Effects of MIO.** Due to the extensive storage of historical tokens in multi-turn retrieval, Table 5 presents the memory cost and mean retrieval performance of various methods. *None* denotes randomly initialized learned tokens $t_N$, *Concat* is concatenating all $t_n^{\mathrm{MIO}}, n \in [1, N-1]$, and *LSTM* and *GRU* respectively indicate using LSTM Graves & Graves (2012) and GRU Dey & Salem (2017) to aggregate all $t_n^{\mathrm{MIO}}$.

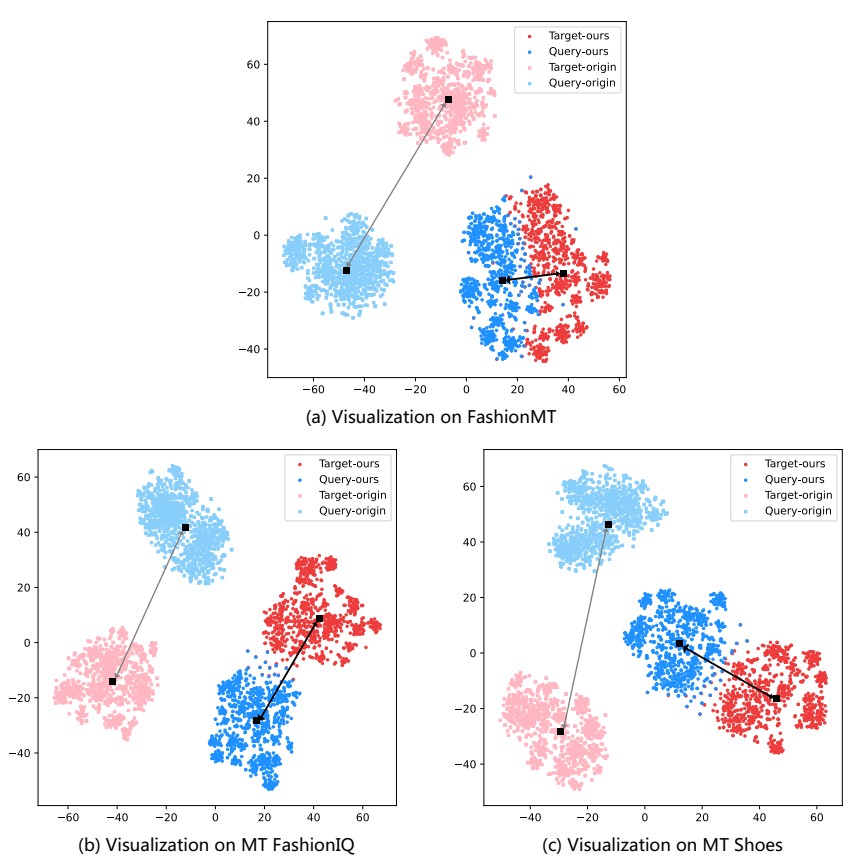

Figure 7: Visualization of modality gaps in FashionMT and existing datasets. Our approach significantly reduces the gap between the query and target sides.

