# OpenReview forum: "MAI: A Multi-turn Aggregation-Iteration Model for Composed Image Retrieval"
_ICLR.cc/2025/Conference — ICLR 2025 Poster_

### Official Review · Reviewer_GvMC · 2024-10-28

**Soundness:** 2
**Presentation:** 3
**Contribution:** 2
**Rating:** 6
**Confidence:** 3

**Summary:**

The paper introduces FashionMT and the MAI model. While the method shows strong results, its complexity and potential scalability challenges could limit its immediate applicability in more generalized settings. Further testing on diverse datasets and longer interaction sequences, along with analysis of the model's efficiency, would enhance the practical relevance of this work.

**Strengths:**

- FashionMT is a significant contribution. The dataset provides richer and more realistic interactions with over 1 million images and 95 categories. This dataset fills a critical gap in the field where existing datasets fail to incorporate historical context across multiple turns.
The model’s approach to maintaining and optimizing key tokens during multiple retrieval iterations is highly relevant to real-world e-commerce scenarios, where users iteratively refine their search queries. The MAI method effectively addresses the shortcomings of the “multiple single-turn” paradigm that fails to leverage historical turn information.
- Extensive experiments show that MAI achieves significant improvements over existing methods in both the combination and rollback settings.

**Weaknesses:**

- While the model is innovative, it introduces considerable complexity with multiple components (TSA, CCL,). The addition of multiple layers, clustering mechanisms, and token filtering might make the model difficult to implement and optimize in real-world settings where computational efficiency is key.
- While the results on FashionMT are strong, the paper does not provide comparisons on non-fashion datasets. The model's application to a more diverse range of image retrieval tasks, such as general object retrieval or scene retrieval, would provide a stronger claim to its versatility.

**Questions:**

- Can the MAI model be adapted to other MTCIR applications outside fashion, such as general e-commerce, furniture, or other products?

- It would be beneficial to know how this impacts performance when dealing with significantly larger datasets. Is the trade-off between memory efficiency and retrieval performance consistent across different dataset sizes?

---

> ### Author Response · Authors · 2024-11-25
>
> # Response to Reviewer GvMC
>
> 1. **Model's Efficiency**
>
>    Thank you for your valuable feedback. We evaluate our approach against existing baselines on three metrics: training time (50 epochs), convergence time to optimal performance, and average inference time per query, as summarized below. All time comparisons are performed on the same device. Two key insights can be drawn from the results:
>
>    (1) While the inclusion of the proposed modules slightly increases inference time, this overhead is **acceptable given the significant performance improvement** (as the CCL loss is not computed during inference).
>
>    (2) Adding TSA, CCL, and MIO components **significantly accelerates convergence to optimal performance** during training. This is attributed to the strong guidance provided by TSA and CCL in the early stages of training, facilitating **efficient fine-tuning and deployment in new scenarios**.
>
>     |Methods|Training Time|Convergence Time|Inference Time|Mean Recall|
>     |-|:-:|:-:|:-:|:-:|
>     |FashionERN|9h 04min|9h 33min|43 ms|38.44|
>     |Image2Sentence|7h 51min|7h 32min|47 ms|49.15|
>     |SPRC|8h 40min|8h 28min|34 ms|58.01|
>     |Base|8h 37min|7h 56min|33 ms|50.04|
>     |**+ MIO**|8h 41min|7h 57min|36 ms|55.60|
>     |**+ MIO + TSA**|8h 58min|7h 21min|41 ms|64.69|
>     |**+ MIO + TSA + CCL (MAI)**|9h 07min|**6h 11min**|41 ms|**66.64**|
>
> 2. **Performance Across Domains and Extended Interactions**
>
>     To validate the effectiveness and generalization capability of the proposed model in more generalized settings, we conduct performance comparisons across three scenarios:
>
>     **(1) CIRR**. Please refer to the **Global Author Rebuttal** for additional experiments, we evaluate performance on the single-turn composed retrieval dataset CIRR, which involves general objects and scenes.
>    **(2) CIRR-MT**. To assess multi-turn composed retrieval in the general domain, we adapt the CIRR dataset using our proposed Modification Generation Framework (MGF) to create CIRR-MT. This dataset contains 30,000 transactions (larger than existing multi-turn datasets) with varying turn numbers (3-turn, 6-turn, and 8-turn). The results are summarized in the table below.
>
>     |Methods|3-turn Recall|6-turn Recall|8-turn Recall|Mean Recall|
>     |-|:-:|:-:|:-:|:-:|
>     |FashionNTM|54.27|51.52|50.88|52.22|
>     |DQU-CIR|57.95|55.13|54.11|55.73|
>     |Context-I2W|56.37|53.47|52.31|54.05|
>     |Image2Sentence|59.42|58.77|57.98|58.72|
>     |SPRC|61.39|59.00|58.02|59.47|
>     |**MAI (ours)**|65.07|64.04|62.31|**63.81**|
>
>    **(3) Product-MT**. To evaluate performance in general e-commerce scenarios, we adapt the large-scale e-commerce dataset M5Product [f], which includes diverse categories such as furniture, toys, etc. Using MGF, we construct the Product-MT test set with the same multi-turn configurations (3-turn, 6-turn, and 8-turn) as CIRR-MT.
>
>     |Methods|3-turn Recall|6-turn Recall|8-turn Recall|Mean Recall|
>     |-|:-:|:-:|:-:|:-:|
>     |FashionNTM|42.45|40.69|38.42|40.52|
>     |DQU-CIR|51.41|49.49|46.09|49.00|
>     |Context-I2W|48.86|46.30|43.02|46.06|
>     |Image2Sentence|50.16|49.42|48.81|49.46|
>     |SPRC|53.46|53.01|51.90|52.79|
>     |**MAI (ours)**|55.10|54.23|52.74|**54.02**|
>
>    From these additional experiments, we conclude that our approach consistently achieves superior retrieval performance in both **generalized domains** and **longer interaction sequences**, demonstrating its robustness and effectiveness.
>
>    [f] Dong et al. M5Product: Self-harmonized Contrastive Learning for E-commercial Multi-modal Pretraining, CVPR 2022.
>
> 3. **Performance on Larger Data Scale**
>
>     To evaluate our approach on larger-scale data and analyze its memory efficiency, the constructed Product-MT test set includes three turn settings, each containing 131,710 transactions, resulting in a total of 395,130 transactions—**16 times larger** than the FashionMT test set in the original paper.
>
>     Following the same settings as Table 5 in the original work, we evaluate the SOTA method SPRC under different module settings to measure mean memory cost and mean recall.
>
>     |Methods|Memory Cost|Mean Recall|
>     |-|:-:|:-:|
>     |SPRC|0 MB|38.88|
>     |+ Concat|384 MB|44.41|
>     |+ LSTM|1341 MB|51.38|
>     |+ GRU|1242 MB|48.06|
>     |+ NTM|1654 MB|52.79|
>     |**MAI (ours)**|64 MB|**54.02**|
>
>     As shown in the table, our approach achieves **optimal performance** with **relatively low memory overhead**.

---

> > ### Comment · Reviewer_GvMC · 2024-12-03
> >
> > Thanks for your detailed explanation. I will keep my positive rating.

---

> > > ### Author Response · Authors · 2024-12-03
> > >
> > > Thank you for your thoughtful feedback. We sincerely appreciate your constructive comments and suggestions, which have been invaluable in improving the quality of our work.

---

### Official Review · Reviewer_JTgq · 2024-11-02

**Soundness:** 3
**Presentation:** 3
**Contribution:** 3
**Rating:** 6
**Confidence:** 3

**Summary:**

This paper introduces MAI (Multi-turn Aggregation-Iteration), a model for multi-turn composed image retrieval (MTCIR), along with FashionMT, a new large-scale dataset specifically designed for MTCIR. The key innovation is a two-stage semantic aggregation approach that uses image captions as a bridge between visual and textual modalities, plus a memory-efficient mechanism for retaining historical information across multiple turns.

**Strengths:**

1. Creation of a large-scale, diverse dataset (FashionMT) that better reflects real-world scenarios
2. Memory-efficient design through the Multi-turn Iterative Optimization mechanism

**Weaknesses:**

1. Limited evaluation of existing datasets (mostly focused on their new dataset), how is the model's performance on existing benchmark like CIRR, FashionIQ and CIRCO.
2. The fixed number of turns (4) in the dataset may not reflect varying real-world scenarios, does the author have plan to extend the dataset with a different number of turns?
3. More visualization of the dataset is expected: I'd like to know (1)sample image sequences across multiple turns: namely the quality of the dataset, the relation of images in different turns, and whether the modified text can describe the relationship between images. (2) visualizations of the distribution of different types of modifications: such as Rollback operations and combination operations.

**Questions:**

1. Statistics for Modification text types: is possible to supply a category of the modified text type, so that the performance can be compared according to different categories? such as Rollback operations and combination operations.
2. Will the dataset be open-source?
3. How does the model perform with varying numbers of turns beyond the fixed 4-turn setup?
4. Could the approach be extended to other domains beyond fashion?

**Details Of Ethics Concerns:**

no need

---

> ### Author Response · Authors · 2024-11-25
>
> # Response to Reviewer JTgq
>
> 1. **Evaluation on Existing Datasets**
>
>    Thank you for your valuable comments. Please refer to the **Global Author Rebuttal** for additional experiments. We have included comparison results across the following three categories: 1. Multi-turn: MT FashionIQ and MT Shoes. 2. Single-turn: FashionIQ, CIRR and CIRCO. The **consistently optimal performance** across these existing datasets demonstrates the effectiveness and generalization of our proposed MAI in multimodal information aggregation for diverse retrieval settings.
>
> 2. **Variable Turn Numbers**
>
>     Thanks to our proposed Modification Generation Framework (MGF), we efficiently construct multi-turn retrieval datasets with **arbitrary turn numbers** for **any image domain**. Specifically, we extend our dataset, FashionMT, to include 3-turn, 6-turn, and 8-turn settings, ensuring an equal number of transactions for each configuration. The results below demonstrate that our approach consistently outperforms across different turn counts.
>
>     |Methods|3-turn Recall|6-turn Recall|8-turn Recall|Mean Recall|
>     |-|:-:|:-:|:-:|:-:|
>     |FashionNTM|36.26|36.42|35.87|36.18|
>     |Context-I2W|45.89|44.71|42.19|44.26|
>     |Image2Sentence|49.15|48.01|47.72|48.29|
>     |SPRC|58.01|57.64|56.33|57.33|
>     |**MAI (ours)**|66.64|66.47|65.98|**66.36**|
>
>     Additionally, we plan to extend our dataset to version 2, **ALL-MT**, leveraging MGF to scale to tens of millions of transactions with variable turn counts, encompassing both e-commerce and general domains, aiming to further drive progress in the multi-turn retrieval field.
>
> 3. **General Domain**
>
>     Please refer to the *Variable Turn Numbers and General Domain* section in **Global Author Rebuttal**, where we use MGF to adapt the general-domain CIRR dataset for multi-turn retrieval, resulting in the CIRR-MT dataset. Our approach achieves leading results on this dataset, demonstrating its robustness across different domains.
>
> 4. **Statistics for Modified Text**
>
>     We categorize modified text based on the following two criteria to compare performance.
>     **(1) Retrospective-based setting**. Table 2 in the main paper categorizes performance into two retrospective-based modified text scenarios: Rollback and Combination. Additional statistics and corresponding mean recall for these two settings are provided below.
>
>     |Settings|Total Count|Proportion|Average Length|Length Variance|Mean Recall|
>     |-|:-:|:-:|:-:|:-:|:-:|
>     |Combination|162,938|65.72%|28.9|4.0|73.34|
>     |Rollback|84,973|34.28%|15.4|3.5|59.94|
>
>     **(2) Product category**. Following the evaluation protocol on the FashionIQ dataset, we provide a comparison of performance across 12 major product categories in the table below.
>
>     |Methods|Dress|Shirts|Skirts|Jacket|Outdoor|Ornament|Pants|Tops|Shoes|Winter|Bags|Underwear|Mean Recall|
>     |-|:-:|:-:|:-:|:-:|:-:|:-:|:-:|:-:|:-:|:-:|:-:|:-:|:-:|
>     |Image2sentence|63.44|58.30|57.10|59.28|58.87|50.78|46.10|60.23|55.67|53.54|45.01|50.12|54.87|
>     |SPRC|66.33|60.40|62.75|59.39|60.28|51.86|49.78|62.85|59.74|57.80|49.14|55.80|58.01|
>     |**MAI (ours)**|72.79|69.34|69.84|68.08|69.02|59.40|60.29|70.27|68.84|67.85|60.60|65.36|**66.64**|
>
> 5. **More Visualization**
>
>     Additional visualizations have been included in Section 7.4 of the Appendix, addressing three key aspects.
>
>     **(1) Dataset Examples**. We provide examples from the proposed FashionMT dataset under both the Combination and Rollback settings to offer better insights into the new dataset. Additionally, to validate the quality of the dataset, Section 7.3 in the Appendix includes a **Quality Control** analysis comparing our proposed dataset with existing multi-turn datasets in terms of accuracy, granularity, and coverage.
>
>     **(2) Dataset Statistics Visualizations**. Visualizations related to dataset statistics are added, including the proportion of the two settings, major category distributions for each setting, text length comparisons, and a dataset scale comparison with existing datasets.
>
>     **(3) Modality Gap**. To validate the theoretical insights presented in Section 4.3, we include t-SNE visualizations of the query-side and target-side representations from the final turn, both with and without the proposed MAI approach, to demonstrate the reduction in modality gap.
>
> 6. **Dataset Availability**
>
>     We will release the FashionMT dataset along with the dataset construction framework MGF to facilitate the progress of multi-turn composed retrieval scenarios.

---

### Official Review · Reviewer_WPSv · 2024-11-03

**Soundness:** 3
**Presentation:** 2
**Contribution:** 2
**Rating:** 5
**Confidence:** 4

**Summary:**

The paper proposes a new dataset, FashionMT, specifically designed for multi-turn composed image retrieval (MTCIR) tasks. FashionMT is characterized by its retrospective-based nature, where the modified text in each new turn may involve information from historical reference images, and its massive diversity, containing 14 times more fashion images and 30 times more categories than previous MTCIR datasets. The authors also introduce a new Multi-turn Aggregation-Iteration (MAI) model that focuses on efficient aggregation and iterative optimization of multimodal semantics in MTCIR. The MAI model includes a Two-stage Semantic Aggregation (TSA) paradigm and a Cyclic Combination Loss (CCL) to enhance semantic consistency and modality alignment, as well as a Multi-turn Iterative Optimization (MIO) mechanism to dynamically select representative tokens and reduce redundancy during multi-turn iterations.

**Strengths:**

a.	The paper proposes a new dataset, FashionMT, for multi-turn composed image retrieval (MTCIR) tasks
b.	FashionMT is characterized by its retrospective-based nature and massive diversity, containing 14 times more fashion images and 30 times more categories than previous MTCIR datasets
c.	The authors introduce a new Multi-turn Aggregation-Iteration (MAI) model that focuses on efficient aggregation and iterative optimization of multimodal semantics in MTCIR
d.	The MAI model includes a Two-stage Semantic Aggregation (TSA) paradigm and a Cyclic Combination Loss (CCL) to enhance semantic consistency and modality alignment
e.	The MAI model also includes a Multi-turn Iterative Optimization (MIO) mechanism to dynamically select representative tokens and reduce redundancy during multi-turn iterations

**Weaknesses:**

a.	How to avoid false negative samples during FashionMT dataset construction? The existing single-turn dataset already has the presence of false negative samples, will it be more obvious with multiple rounds?.
b.	For the ZS-CIR method, how was it trained and tested on FashionMT? They are designed for zero-shot, is the adaptation process also zero-shot paradigm?
c.	How does MAI perform for the single-turn dataset? The authors should add the results of MAI for datasets such as FashionIQ, CIRR, etc. to confirm the superiority of MAI even with turn=1.
d.	The clustering used in MIO is affected by the number of cluster centers, denoted as k, and the authors do not mention in the implementation details the value of k, and the number of iterations used when performing k-means, which affects the accuracy of the clustering. And does clustering increase the time overhead of the whole model? Authors need to increase the comparison of model training and inference time.
e.	The Q-former used in MAI is initialized by BLIP-2 with the Flan-t5-xxl language model, while the version of BLIP-2 used in SPRC is blip2-pretrain, right? If MAI uses the same version of BLIP-2 as SPRC to initialize the Q-former parameters, how does the result behave? The authors need to add this experiment to further demonstrate the superiority of MAI?
f.Typos and minors. There are non-standard punctuation marks, such as Line 431, P8, "thanks to TSA", etc. It is recommended to check the whole paper.

**Questions:**

refer to the weaknesses.

---

> ### Author Response · Authors · 2024-11-25
>
> # Response to Reviewer WPSv
>
> 1. **False Negative Samples**
>
>     Thank you for your insightful question. We argue that the presence of false negative samples in existing single-turn datasets primarily arises from two factors: (1) ignoring the possibility of multiple similar targets, and (2) overly simplistic modified text descriptions. To address these issues, we propose the following solutions.
>
>     **(1) Target Similarity Filtering**. Based on the caption generated from image $I$, we utilize a pre-trained CLIP model to measure the similarity between the caption text embedding and image embeddings. We filter out images other than $I$ that exceed a set similarity threshold. If the number of filtered images exceeds 20, it indicates a high risk of false negatives for this target, and we discard image $I$ to avoid potential issues.
>
>     **(2) Fine-grained Attribute Modified Text**. By introducing fine-grained attributes, our generated modified texts are more precise and detailed, with an average length approximately 2.4 times longer than existing single-turn datasets like FashionIQ. This enhanced specificity allows for more accurate target localization.
>
>     Given the significance of this issue for our task, we also conduct a quality evaluation in Appendix Section 7.3 (Table 10) using metrics such as Accuracy, Granularity, and Coverage, highlighting notable improvements over existing benchmarks.
>
> 2. **ZS-CIR Methods**
>
>     While ZS-CIR methods are tailored for zero-shot tasks, their multimodal semantic fusion paradigm serves as a strong baseline due to its ability to unify representations in the text space, which can be advantageous in text-rich multi-turn interaction scenarios.
>
>     **(1) Training**. The reference image is mapped into tokens via a mapping network, concatenated with the modified text, and then processed by a text encoder. Multi-turn features are aggregated using the FashionNTM method and aligned with target image features. Following the training configurations of these ZS-CIR methods, we train only the mapping network for Pic2word and Context-I2W, while for Image2sentence, both the image encoder and mapping network are optimized.
>
>     **(2) Testing**. All modules are frozen, and similar to the training process, we obtain aggregated multi-turn features, which are then compared with image features from the gallery to calculate similarity for retrieval. This training strategy yields better performance compared to direct zero-shot testing, with some methods achieving competitive results.
>
> 3. **Performance on Single-turn Datasets**
>
>     Please refer to the **Global Author Rebuttal** for additional experiments on single-turn datasets, FashionIQ and CIRR. Our approach also achieves SOTA results compared to existing methods, validating its effectiveness in the specific case where the number of turns is 1.
>
> 4. **Clustering Details**
>
>     In the MIO module, the number of cluster centers (`n_cluster`) is set to 32, balancing performance and efficiency, as shown in the table below.
>
>     |Number|Inference Time|Mean Recall|
>     |-|:-:|:-:|
>     |16|35 ms|55.11|
>     |32|36 ms|55.60|
>     |64|39 ms|55.64|
>
>     Unlike the GMM algorithm, which relies on an iterative process (typically Expectation-Maximization) to estimate optimal parameters such as means, covariances, and mixture weights, our approach **eliminates the need for iterative updates**. As a non-parametric, instance-based approach, it directly utilizes the training data to compute distances between query points and stored data during inference, avoiding iterative optimization or parameter fitting.
>
>     Additionally, we analyze its impact on training (50 epochs) and inference time per query, comparing with two parameter-free baselines. The results show that **MIO introduces minimal overhead** while selectively retaining critical information, **enhancing performance** and **reducing memory usage**.
>
>     |Methods|Training Time|Inference Time|Mean Recall|
>     |-|:-:|:-:|:-:|
>     |Base|8h 37min|33 ms|50.04|
>     |+ DBSCAN|8h 39min|34 ms|54.28|
>     |+ GMM|20h 11min|117 ms|54.97|
>     |+ **MIO (ours)**|8h 41min|36 ms|55.60|
>
> 5. **Backbone**
>
>     BLIP-2 with the Flan-t5-xxl language model is the image captioning model we use for caption extraction. To ensure a fair comparison with the existing SOTA method SPRC, we initialize the Q-Former parameters using the same **blip2_pretrain_vitL** model as SPRC in all current experiments, including the base model in Table 5. This detail has been added to the Implementation Details section.
>
> 6. **Typos**
>
>     Thank you for pointing out the typos and minor issues. We have thoroughly reviewed the entire paper and corrected the non-standard punctuation marks, including the ones you mentioned.

---

### Official Review · Reviewer_TgqM · 2024-11-04

**Soundness:** 4
**Presentation:** 4
**Contribution:** 4
**Rating:** 8
**Confidence:** 5

**Summary:**

Existing multi-turn composed image retrieval (MTCIR) methods adopt the “multiple single-turn” paradigm and neglect the historical correlation information in multi-turn interactions. To address this problem, this paper first builds a new retrospective-based MTCIR dataset, wherein modification demands are highly associated with historical turns. Then, the authors develop a new Multi-turn Aggregation-Iteration (MAI) model, which contains a two-stage semantic aggregation paradigm coupled with a cyclic combination loss. Besides, a multi-turn iterative optimization mechanism is designed to select representative tokens and reduce redundancy dynamically. Experimental results demonstrate the effectiveness of the proposed method.

**Strengths:**

1)	This paper focuses on an interesting multi-turn composed image retrieval task, and points out the critical “multiple single-turn” problem in the existing research field. This research is very valuable and provides significant insights for further research.
2)	The authors have constructed a new dataset for the multi-turn composed image retrieval task, which is more similar to real-world scenarios, and is more massive and diverse. This is an important contribution if the dataset is made publicly available.
3)	A multi-turn key information-aware approach, named the Multi-turn Aggregation model, is proposed to achieve multimodal semantics aggregation and multi-turn information optimization. The proposed method is reasonable and the parameter-free multi-turn iterative optimization (MIO) mechanism is interesting.
4)	The effectiveness of the proposed method has been demonstrated by extensive experimental results.
5)	The writing of this paper is good and easy to follow.

**Weaknesses:**

1)	Will the collected dataset be made publicly available? Some key characteristics of the dataset are still not clear, such as the average and variance of turn number for each query, the average length for modification text, etc. The dataset will be more comprehensible as more detailed statistics are provided.
2)	It seems MIO is executed for each input, and it contains a DPC-kNN for clustering and density computation. The complexity of this module should be analyzed.
3)	In Table 6, what does the w/o CCL mean? It seems w/o CCL has outperformed most of the compared methods. Are all the methods evaluated on the same backbone? If they are not, the comparison results may not be fair.
4)	Except for the proposed dataset, the proposed method should be also evaluated on existing multi-turn composed image retrieval datasets to further demonstrate the effectiveness.

**Questions:**

please try to address the weaknesses.

---

> ### Author Response · Authors · 2024-11-25
>
> # Response to Reviewer TgqM
>
> 1. **Dataset Availability and Key Characteristics**
>
>     Thank you for your valuable comments. We will release our FashionMT dataset along with the dataset construction framework MGF to support the advancement of multi-turn composed retrieval scenarios.
>     Additionally, we provide more details on the key characteristics of the dataset in the table below. For consistent performance comparison across methods, our FashionMT dataset currently fixes the number of turns per transaction to 3.
>
>     |Metrics|Value|Metrics|Value|
>     |-|:-:|-|:-:|
>     |# Images|1,067,688|# Transactions|247,911|
>     |# Turns|743,733|# Categories|95|
>     |# Attributes|3782|Avg Attribute per Query|3.72|
>     |Avg Mod Length|24.3|Mod Length Variance|3.8|
>     |Avg Caption Length|18.2|Caption Length Variance|4.5|
>
> 2. **Complexity of MIO**
>
>     The complexity of the proposed MIO module is analyzed from two perspectives: **algorithmic complexity** and **experimental response time**. We also replace DPC-kNN with two representative parameter-free algorithms, DBSCAN and Gaussian Mixture Models (GMM).
>
>     (1) **Algorithmic Complexity Analysis.**
>     $N$ represents the number of tokens (e.g. 64), and $k$ is the number of selected tokens (e.g. 32). $B$ is the batch size (e.g. 2048), and $C$ is the token dimension (e.g. 256).
>
>       * MIO Module Complexity:
>         * Distance Computation: $\mathcal{O}(B \times N \times C)$
>         * Density Estimation: $\mathcal{O}(B \times N \times C)$
>         * Token Pruning: $\mathcal{O}(B \times k \times C)$
>         * **Total**: $\mathcal{O}(B \times (N + k) \times C) \approx \mathcal{O}(B \times N \times C)$
>       * DBSCAN Complexity: $\mathcal{O}(B \times N \times C)$
>       * GMM Complexity per iteration: $\mathcal{O}(B \times (N \times k + k^2) \times C)$
>
>     In summary, the proposed MIO module achieves **computational efficiency** by (1) Eliminating the quadratic scaling and iterative procedures typical of GMM, offering advantages with small $N$ and moderate $k$. (2) Employing learnable tokens for multi-turn embeddings with a reduced dimension of 256, which is lower than the original embedding dimension, leading to reduced time and memory overhead.
>
>     (2) **Experimental Complexity Analysis**.
>     The table below presents the average response time per sample and the corresponding recall values on FashionMT's test set. Our proposed MIO demonstrates advantages in **runtime efficiency and performance**.
>
>     |Methods|Inference Time|Mean Recall|
>     |-|:-:|:-:|
>     |DBSCAN|34 ms|54.28|
>     |GMM|117 ms|54.97|
>     |**MIO (ours)**|36 ms|55.60|
>
> 3. **Ablation of Components in CCL Loss**
>
>     Table 6 presents ablation experiments to further validate the effectiveness of our proposed two-stage fusion process. When **w/o CCL** (i.e., without CCL), the results reflect the **performance of only the TSA module**. Due to the TSA module's effectiveness in multimodal semantic aggregation, its performance surpasses methods like FashionERN. The addition of the CCL constraint further enhances performance. All compared methods utilize the **same ViT-L backbone** to ensure a fair comparison.
>     For better readability, we have also reorganized Table 6 as follows.
>
>     |$L(t_n^r, t_n^{tg})$|$L(t_n^{tg}, m_n)$|$L(m_n, c_n^{tg})$|$L(c_n^{tg}, t_n^r)$|Mean Recall|
>     |:-:|:-:|:-:|:-:|-|
>     |✓|✓|✓|✓|65.57 (TSA + CCL)|
>     |×|✓|✓|✓|63.54|
>     |✓|×|✓|✓|64.39|
>     |✓|✓|×|✓|64.51|
>     |✓|✓|✓|×|64.90|
>     |×|×|×|×|62.48 (TSA)|
>
> 4. **Evaluation on Existing Datasets**
>
>    Please refer to the **Global Author Rebuttal** for additional experiments. Through performance comparisons on existing multi-turn composed retrieval datasets, MT FashionIQ and MT Shoes, as well as single-turn datasets, our proposed MAI consistently achieves optimal results, demonstrating both its effectiveness and generalizability.

---

> > ### Comment · Reviewer_TgqM · 2024-11-26
> >
> > Thanks for the detailed response from the authors. These responses have fully addressed my questions, therefore, I will keep the rating of acceptance ~

---

> > > ### Author Response · Authors · 2024-11-26
> > >
> > > Thank you so much for your reply. We sincerely appreciate your constructive comments and suggestions, which greatly enhance the quality of our paper.

---

### Author Response · Authors · 2024-11-25

# Global Author Rebuttal

We would like to express our sincere gratitude for the insightful and helpful feedback provided by the reviewers, especially the encouraging comments: **valuable dataset (Reviewer TgqM, WPSv, JTgq, GvMC)**, **interesting and novel approach (TgqM, GvMC)**, **valuable insights (TgqM)**, **significant improvements (GvMC)**, and **easy to understand (TgqM)**.

In the following separate response section, we will address each reviewer's comments in detail and provide responses to enhance the quality of our work.

Before addressing all questions, please allow us to highlight two key aspects of interest: **(1) evaluation on additional datasets** and **(2) dataset availability**, to further clarify and emphasize the contributions of this study.

1. **Evaluation on Additional Datasets**

    **(1) Multi-turn Composed Retrieval.**
    Despite limitations in existing multi-turn composed retrieval datasets, we add mean recall comparisons on the existing benchmarks MT FIQ and MT Shoes. The table below shows that our approach achieves the **best performance** in all settings due to its fine-grained semantic capture and efficient modality alignment.

    |Methods|MT FashionIQ|MT Shoes|FashionMT (ours)|Mean Recall|
    |-|:-:|:-:|:-:|:-:|
    |FashionNTM|48.1|31.2|36.26|38.52|
    |DQU-CIR|55.9|46.2|39.88|47.33|
    |Context-I2W|49.1|36.5|45.89|43.83|
    |Image2Sentence|51.1|47.0|49.15|49.08|
    |SPRC|57.4|52.7|58.01|56.04|
    |**MAI (ours)**|62.2|56.7|66.64|**61.85**|

    **(2) Single-turn Composed Retrieval.** We follow the standard single-turn composed retrieval settings. For CIRR and FashionIQ, models are independently trained on their respective datasets and evaluated on the test sets for average recall, denoted as Avg. For CIRCO, due to the absence of a training set, we adopt the approach from CoVR2 [a], training on CC-CoIR and testing on the CIRCO test set. By effectively aggregating fine-grained multimodal semantics in composed retrieval scenarios, our approach demonstrates competitive performance even in the single-turn setting.
    * **CIRR and FashionIQ:**

        |Methods|CIRR|FashionIQ|Mean Recall|
        |-|:-:|:-:|:-:|
        |CoVR2 [a]|78.92|60.57|69.75|
        |CALA [b]|78.74|57.96|68.35|
        |ECDE [c]|78.09|60.44|69.27|
        |FashionERN|74.85|62.62|68.74|
        |DQU-CIR|74.55|65.13|69.84|
        |SPRC|82.66|64.85|73.76|
        |**MAI (ours)**|86.44|68.12|**77.28**|

        [a] Ventura et al. Covr-2: Automatic data construction for composed video retrieval, TPAMI 2024.
        [b] Jiang et al. Cala: Complementary association learning for augmenting comoposed image retrieval, SIGIR 2024.
        [c] Thawakar et al. Composed video retrieval via enriched context and discriminative embeddings, CVPR 2024.

    * **CIRCO:**

        |Methods|mAP@5|mAP@10|mAP@25|Mean Recall|
        |-|:-:|:-:|:-:|:-:|
        |Pic2word|8.72|9.51|10.46|9.56|
        |LinCIR [d]|12.59|13.58|15.00|13.72|
        |Image2Sentence|13.19|13.83|15.20|14.07|
        |CoVR [e]|21.43|22.33|24.47|22.74|
        |CoVR2|23.50|24.66|27.05|25.07|
        |**MAI (ours)**|25.18|25.91|27.22|**26.10**|

        [d] Gu et al. Language-only training of zero-shot composed image retrieval, CVPR 2024.
        [e] Ventura et al. CoVR: Learning composed video retrieval from web video captions, AAAI 2024.

    **(3) Variable Turn Numbers and General Domain.** To evaluate multi-turn composed retrieval in the general domain, we employ the proposed Modification Generation Framework (MGF) to transform the CIRR dataset, which includes general objects and scenes, into CIRR-MT. This dataset consists of 30,000 transactions with varying turn counts (3-turn, 6-turn, and 8-turn), exceeding the scale of existing multi-turn datasets. The results are presented in the table below.

    |Methods|3-turn Recall|6-turn Recall|8-turn Recall|Mean Recall|
    |-|:-:|:-:|:-:|:-:|
    |FashionNTM|54.27|51.52|50.88|52.22|
    |DQU-CIR|57.95|55.13|54.11|55.73|
    |Context-I2W|56.37|53.47|52.31|54.05|
    |Image2Sentence|59.42|58.77|57.98|58.72|
    |SPRC|61.39|59.00|58.02|59.47|
    |**MAI (ours)**|65.07|64.04|62.31|**63.81**|

2. **Dataset Availability**

    We will release our dataset through a dedicated website upon acceptance of the paper. The website will include:
    * **Image Download Links**. Providing access to all images in the FashionMT dataset.
    * **Related Text**. Including the modified text and captions for the images.
    * **Documentation**. Covering data format, usage guidelines, and annotations.
    * **Terms of Use**. Outlining usage policies to ensure ethical and appropriate use of the dataset.
    * **Modification Generation Framework (MGF)**. We plan to release the proposed MGF, compatible with various LLMs (e.g., Llama-3.1-8B-Instruct and Qwen2.5-7B), facilitating the creation of tailored multi-turn scenarios.

---

### Meta-Review · Area_Chair_BcfS · 2024-12-18

**Metareview:**

This paper was reviewed by four experts in the field. The final ratings are 8,6,5,6. This paper studies the problem of Multi-Turn Composed Image Retrieval (MTCIR). It proposes a new dataset with the aim of addressing the limitation of existing methods which does not incorporate historical information. It also proposes a Multi-turn Aggregation-Iteration based on the new dataset.

Reviewers agree that this paper addresses an interesting problem. The proposed dataset is likely to be a valuable contribution. The proposed MAI model successfully addresses the limitation of previous methods. Overall, this paper is acceptable to ICLR.

**Additional Comments On Reviewer Discussion:**

Reviewers WPSv and JTgq did not response to authors' rebuttal. The AC has take this into account and consider the rebuttal reasonably addressed the reviewers' concerns.

---

### Decision · Program_Chairs · 2025-01-22

Accept (Poster)